# StableSemantics: A Synthetic Language-Vision Dataset of Dense Semantic Representations in Naturalistic Images

## Abstract

Understanding dense visual semantics remains a fundamental challenge in computer vision, as semantically similar objects can exhibit drastically different visual appearances. Recent advancements in generative text-to-image frameworks have led to models that implicitly capture natural scene statistics. These models learn to model complex relationships between objects, lighting, and other visual factors, enabling the generation of detailed and contextually rich images from text captions. To advance visual semantic understanding and develop more robust and interpretable vision models, we present **StableSemantics**, a large-scale dataset composed of 224 thousand human-curated prompts, processed natural language captions, over 2 million synthetic images, and 10 million attention maps. The dataset provides fine-grained semantic attributions at the noun-chunk level, leverages human-generated prompts that correspond to visually interesting stable diffusion generations, and provides 10 generations per phrase, with cross-attention maps corresponding to noun chunks for each image. We explore the semantic distribution of generated images, examine the distribution of objects within images, and benchmark captioning and open vocabulary segmentation methods on our data. As the first diffusion dataset to include dense attention attributions, we expect StableSemantics to catalyze advances in visual semantic understanding and provide a foundation for developing more sophisticated and effective visual models.

## 1 Introduction

Dense visual scene understanding is a complex task that requires the integration of cues, context, and prior knowledge to navigate the inherent variability and complexity of the visual world. This complexity is particularly evident when considering the diversity of visual appearances that can correspond to a single semantic concept. For instance, entities that correspond to "man-made structures" can have vastly different visual appearances, ranging from sleek skyscrapers to rustic cottages. Similarly, objects that serve the same purpose, such as "containers," can have diverse shapes, sizes, and materials. This disconnect between semantic meaning and visual appearance poses a significant challenge for computer vision systems (Brust & Denzler, 2018; Duan & Kuo, 2021; Alqasrawi, 2016; Barz & Denzler, 2020), requiring the disentanglement of the underlying semantic structure from visual differences (Caron et al., 2021; Xu et al., 2023; Elharrouss et al., 2021; VS et al., 2024; Hu et al., 2023; Quinn et al., 2017). To overcome this challenge, recent advances have adopted data-driven approaches, which learn to recognize patterns and relationships in large datasets of images and annotations. However, the reliance on large datasets of images and annotations poses a significant challenge in the development of segmentation models. Acquiring and annotating such datasets can be a time-consuming and resource-intensive process, requiring careful consideration of data quality and diversity.

This limitation has sparked interest in exploring alternative approaches that can reduce the need for large human-annotated datasets. One promising direction is the use of generative models, which have shown impressive results in translating between semantic meaning and visual appearance (Rombach et al., 2022; Podell et al., 2023; Song et al., 2020; Ho et al., 2022). In particular, diffusion-based text-to-image synthesis models have demonstrated an impressive ability to generate highly realistic images from textual descriptions, suggesting that these models must possess an implicit understanding of the

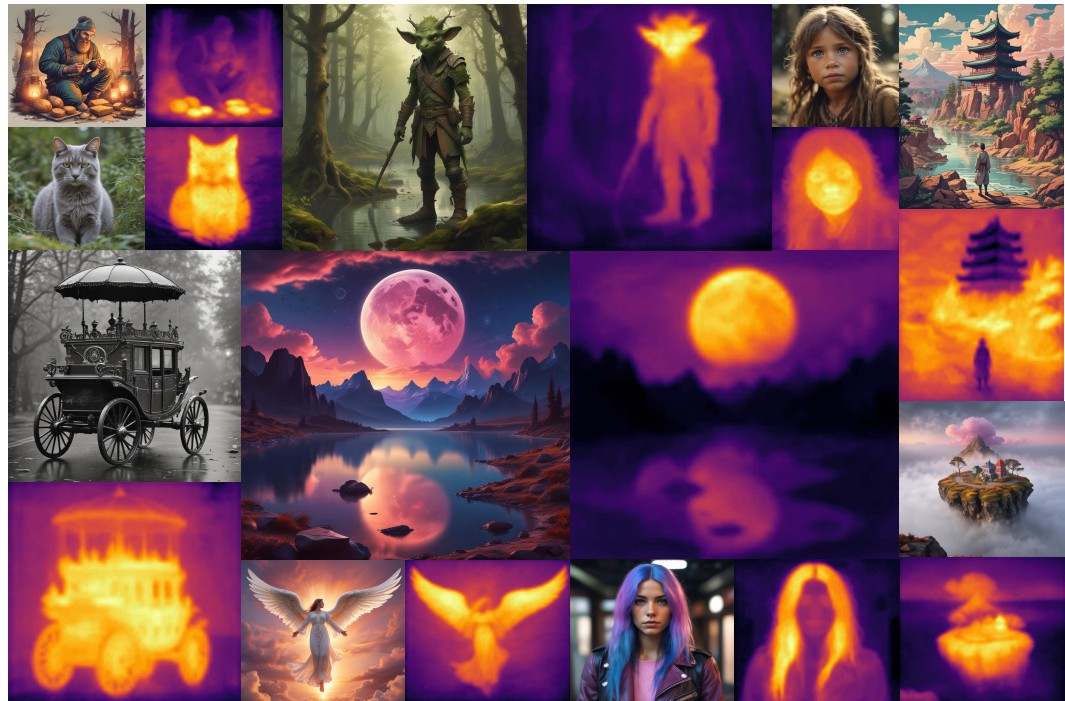

Figure 1: **Images and maps corresponding to select noun chunks from StableSemantics.** Images are generated using natural language captions derived from human generated and curated prompts. For reproducibility, seeds are recorded for each generation. Noun chunks are extracted by performing dependency parsing the natural language captions. Semantic maps corresponding to each noun chunk is computed using the cross-attention maps with the DAAM Tang et al. (2022) method. Only a single attention map is shown here for each image, please see below for additional examples. Yellow indicates high relevance, black indicates low relevance.

semantic structure of the visual world, and have learned to associate words and phrases with specific visual concepts. By leveraging cross-attention mechanisms, these models learn to link textual input to visual representations and enable the generation of images that are grounded in the semantic content of the input text (Tang et al., 2022). This in turn has led to generative media proliferating over social media, advertisements, and news media. Although generated images may seem indistinguishable from real images to the human eye, neural networks can still differentiate between them (Huang et al., 2024; Aziz et al., 2024; You et al., 2024). This suggests that generated images are still quite different from real images and that generative models are still far from perfect in accurately replicating the real data distribution (You et al., 2024). This demonstrates that a very clear gap still exists between generated and real data. Thus, there is a strong need for the development of more advanced generated data to help models perform well on synthetically generated images and to address the difficulty of gathering large-scale real data. Specifically, there are no existing large-scale generated datasets that contribute semantic maps in addition to images and their corresponding captions.

In this work, we introduce **StableSemantics**, a dataset that consists of human-generated and curated prompts, natural language captions, images generated from the captions, and attention attribution maps corresponding to objects in the captions. Unlike prior work which sourced unfiltered human-generated prompts, we source our prompts from a pool of images that have been evaluated by humans for their visual appeal and interest, resulting in a dataset that mirrors the types of images people find engaging. As the original prompts may not always reflect natural language, we employ a large language model to paraphrase and refine them into fluent and natural-sounding captions, thereby bridging the gap between human-generated prompts filtered for visual appeal and naturalistic language. Each natural language caption is provided to a Stable Diffusion XL model to generate high-resolution and reproducible images. Finally, we explicitly record the dense text-to-image cross-attention maps used to condition the image generation process. We visualize the distribution of semantics across images, evaluate the spatial distribution of semantic classes within images, and evaluate the alignment

of current captioning and open-set segmentation models on our dataset. To our knowledge, our dataset is the first to systematically record the spatial distribution of cross-attention activations corresponding to individual noun chunks. **StableSemantics** can be utilized in future research on various vision tasks such as object detection, semantic segmentation, semantically meaningful representation learning, image-inpainting and object removal, object editing, etc.

## 2 RELATED WORK

**Natural Scene Statistics.** Natural image statistics have been a long-standing area of research in computer vision and neuroscience. The human visual system is thought to be adapted to the statistical properties of natural images, which are characterized by complex dependencies between pixels (Girshick et al., 2011; van der Schaaf & van Hateren, 1996). The power law distribution of gradient magnitude statistics is thought to be a result of the hierarchical, self-similar structure of natural images, which arises from the presence of edges, textures, and other features at multiple scales. Understanding natural image statistics has important implications for image recognition tasks (Zoran, 2013; Heiler & Schnörr, 2005; Fang et al., 2012), and has inspired the development of a range of algorithms and models that are tailored to the statistical properties of natural images (Mechrez et al., 2019; Kleinlein et al., 2022; Hepburn et al., 2023; Xiang et al., 2024; Hepburn et al., 2021; Talbot et al., 2023). Other work has also explored the semantic structure of visual data, seeking to understand how higher-level categories and concepts are reflected in the statistical patterns present in images. This work has shown that different categories of images, such as scenes and objects, exhibit distinct statistical patterns (Torralba & Oliva, 2003; Henderson et al., 2023). These semantic statistics have important implications for the development of models that can effectively represent and analyze visual data.

**Deep Image Generative Models.** Recent progress on generative models has enabled the generation of images, video, text, and audio (Podell et al., 2023; Song et al., 2020; Ho et al., 2022; Gupta et al., 2023; Touvron et al., 2023; Evans et al., 2024). Models rely on a variety of different mathematical assumptions and architectures. Variational autoencoders (Kingma & Welling, 2013; Luhman & Luhman, 2023; Harvey et al., 2021; Razavi et al., 2019; Van Den Oord et al., 2017) and flow-based models (Rezende & Mohamed, 2015; Tong et al., 2023; Dinh et al., 2014; Kingma & Dhariwal, 2018), while highly efficient, tend to produce lower-quality samples. Generative Adversarial Networks (GANs) can yield high-fidelity samples but may neglect modes in the data and can exhibit unstable training dynamics (Goodfellow et al., 2014; Karras et al., 2019; Brock et al., 2018; Mirza & Osindero, 2014; Zawar et al., 2022). Auto-regressive methods (Huang et al., 2023; Parmar et al., 2018; Lee et al., 2022; Ramesh et al., 2022), although capable of producing high-quality samples, typically experience slow sampling. Recent progress in energy/score/diffusion models has given us methods that are simultaneously stable during training and yield high-quality samples (Rombach et al., 2022; Ramesh et al., 2022).

**Visual Datasets.** Deep learning models have achieved remarkable results by leveraging vast amounts of data. There has been a significant push to collect large-scale datasets. Earlier works such as LAION-5B (Schuhmann et al., 2022), Flickr Caption (Young et al., 2014), RedCaps (Desai et al., 2021) and YFCC100M (Thomee et al., 2016) scrape real-world data of image-caption pairs from web sources. COCO (Lin et al., 2014) goes a step further to also provide pixel-level segmentation masks on top of the image-caption pairs. (Agrawal et al., 2015; Goyal et al., 2016; Marino et al., 2019; Wang et al., 2018; Krishna et al., 2017) introduce datasets specifically for the task of VQA. Given the difficulty of collecting real data, recently there has been a shift towards synthetic datasets. StableRep (Tian et al., 2024) and Hammoud et al. (2024) also demonstrated the usefulness of Stable Diffusion images in training contrastive image models. Pick-a-Pic (Kirstain et al., 2023) provides a dataset of image-caption pairs where each sample contains a pair of diffusion-generated images and the human preference between those images. JourneyDB (Sun et al., 2023) and DiffusionDB (Wang et al., 2022b) are the closest works to ours that release large-scale datasets of synthetic image-caption pairs.

## 3 DATA COLLECTION

In this section, we provide details on the collection and creation process of our dataset. Our data originates from human-generated and curated prompts submitted publicly by users online for Stable Diffusion XL. We describe our prompt collection process in section 3.1. The prompts are filtered and transformed into natural language captions, and we describe our procedure in section 3.2. Finally, we generate images and compute noun-chunk to image saliency maps via cross-attention in section 3.3.

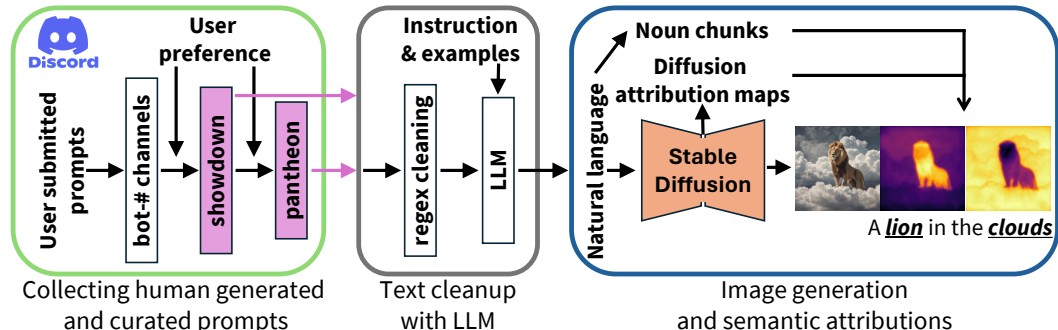

Figure 2: **Data collection and generation process**. **(1)** We collect our data from Stable Diffusion Discord, specifically the *showdown* and *pantheon* channels which are derived from user rankings of images generated from public prompt submissions. **(2)** The prompts are cleaned using regex to remove common errors, and further processed using an LLM to generate natural language captions. **(3)** The natural language captions are provided to a Stable Diffusion XL model, while we record the attention attribution maps corresponding to noun chunks.

### 3.1 COLLECTING HUMAN CURATED PROMPTS

Our dataset is collected from the ***Stable Diffusion*** discord server, where users can publicly submit prompts to generate images using a discord bot. After users submitted prompts using the /dream command, the bot would return images corresponding to a prompt. Beyond accepting a prompt, users could also submit negative prompts, and image styles which were achieved via a prefix/affix pattern of text to the original prompt. These style patterns were not visible to the users.

We started our data collection after the Stable Diffusion XL 1.0 (Podell et al., 2023) candidate was made available via bots. The data was continuously collected from **July 11, 2023** (a day after SDXL 1.0 candidate bots were launched) until **Feb 07, 2024** (SDXL bot shutdown). Users were allowed to submit prompts to bot-# channels where # corresponds to a number. We observed that the number of channels varied over time, and generally remained at slightly over 10. For each prompt, the bot would return 2 images. Users were asked to select which image was better by clicking on a button corresponding to an image, without explicit guidance on what "better" meant. Our understanding from discussions with members of staff was that these prompts and image pairs were used for fine-tuning the SDXL candidates using RLHF/DPO (Ouyang et al., 2022; Rafailov et al., 2024), selection of model candidates, and selection of generation hyperparameters.

Prior work has also collected user-generated prompts from discord servers for MidJourney and Stable Diffusion (Sun et al., 2023; Wang et al., 2022b). Our work goes further by only collecting prompts that were human-curated. The Stable Diffusion discord followed a three-tier hierarchy for prompts, where users first submit and rate images in bot-#, with highly rated images from all bot channels going into a single showdown channel every 15 minutes. The showdown channel was reset every 30 minutes and had the history wiped. In the showdown channel, 2 images and their respective prompts were placed side-by-side. Users again were asked to select the images that were more visually appealing. Every 30 minutes, the top-ranked images and prompts would go into the pantheon channel. The pantheon had history going back to inception May 02, 2023. We note that strictly speaking the showdown to pantheon selection process was not fair to images that came in at the second 15 minute slice, as they were given less time to be voted upon. Due to this, we do not further distinguish between prompts collected from these two sources. Our data collection process ran every 14 minutes on the showdown channels and ran once on the pantheon channel. This was sufficient as after the initial collection date, new pantheon entries were a strict subset of showdown prompts. Visual inspection of the generations from showdown and pantheon suggest that these images were generally more artistic and contained more interesting visual compositions than the bot-# channel. We collect a total of 235k unique user-generated prompts, which is further filtered according to NSFW ratings and caption length.

| Dataset | Total Images | Total Captions | Caption Source | Human Preferred | Open-set Semantics |
|---|---|---|---|---|---|
| COCO 2017 Lin et al. (2014) | 123k | 617k | H | N | N |
| LAION-COCO Schuhmann et al. (2022) | 600M | 600M | M | N | N |
| DiffusionDB Wang et al. (2022b) | 14M | 1.8M | H | N | N |
| JourneyDB Sun et al. (2023) | 4.7M | 1.7M | H+M | N | N |
| **StableSemantics** | 2M | 224k | H+M | Y | 10.8M |

Table 1: **Size of the different components of StableSemantics.** Our captions are selected by humans to correspond to visually interesting images. We are the only dataset to provide dense open-set spatial semantic maps. Our maps are derived from the cross-attention maps in Stable prompts. Note that 235k unique captions are collected, 224k remain after NSFW filtering and only 200k captions are used for image generation after filtering for length.

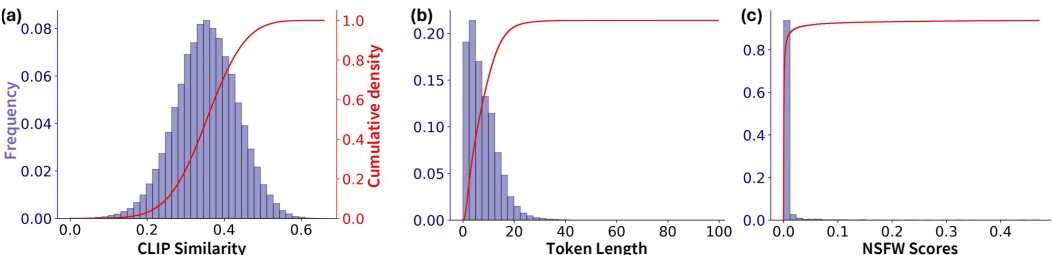

Figure 3: **Histogram of dataset statistics. (a)** We visualize the cosine CLIP similarities between generated images and original captions. **(b)** Number of tokens in the captions. **(c)** NSFW scores of the captions after LLM filtering. Scores measured by LLaMA Guard 2 for sexuality and hate.

## 3.2 OBTAINING NATURAL LANGUAGE CAPTIONS

As shown in Figure 4, the user-submitted prompts generally took a tag-like format, with descriptors being separated by commas. Such prompts are convenient for users to specify and likely achieve good results due to the use of CLIP text networks for conditioning, which can operate like bag-of-words models (Thrush et al., 2022; Yuksekgonul et al., 2022). However, such prompts generally perform poorly when typical NLP pipelines are used for analysis. These prompts further may not explicitly specify needed visual relationships in the text, and instead excessively rely on the prior learned by the diffusion model to disambiguate relationships. In order to mitigate this issue, we utilize an LLM model to clean up the original raw user-generated prompts. We use `Gemini 1.0 Pro` for this task, as it performed competitively against other models at the time of our work (Team et al., 2023) and offered a free API. The model was instructed to take the user-generated prompts and transform them into natural language captions. To enhance the results, we augment the prompt via in-context learning from GPT-4 input/output pairs. To remove NSFW prompts, we record the Gemini API safety ratings for each input prompt, and remove the prompts where a 4 out of 4 rating was given on the axes of sexuality/hate speech/harassment, or if the model itself produced a refusal, or if the prompt was repeatedly returned with an error (blocked by Google). Please see Figure 2 for a visualization of the pipeline.

## 3.3 IMAGE GENERATION AND SEMANTIC ATTRIBUTION

To provide a fully reproducible pipeline for the images and maximize the usability of our dataset, we generate the images ourselves using open weights and record the random seed for each generation. Images are generated using `sdxl lightning 4step unet` (Lin et al., 2024), a few-step distilled version of Stable Diffusion XL. For each prompt, we perform parsing using spaCy `en_core_web_lg` to extract noun chunks. To obtain mappings from noun chunks to spatial attributions, we use Diffusion Attentive Attribution Maps which measures the cross-attention from tokens in the language condition to the UNet. Specifically, we used the improved DAAM-i2i guided heatmap variant (Tang et al., 2022; Chowdhury, 2024) which improves object localization. We observe that unrelated articles like "a", and "the" and possessive determiners like "his", "her", "our", "their" are not typically localized to a specific object, but rather have attribution maps diffuse over the background or various objects in the image. While similar phenomena has been noted in ViTs (Darcet et al., 2023) and pure text LLMs (Clark et al., 2019; Kovaleva et al., 2019; Xiao et al., 2023), our observation is novel in that text-to-image diffusion models are encoding contextual information in these "filler" words. For

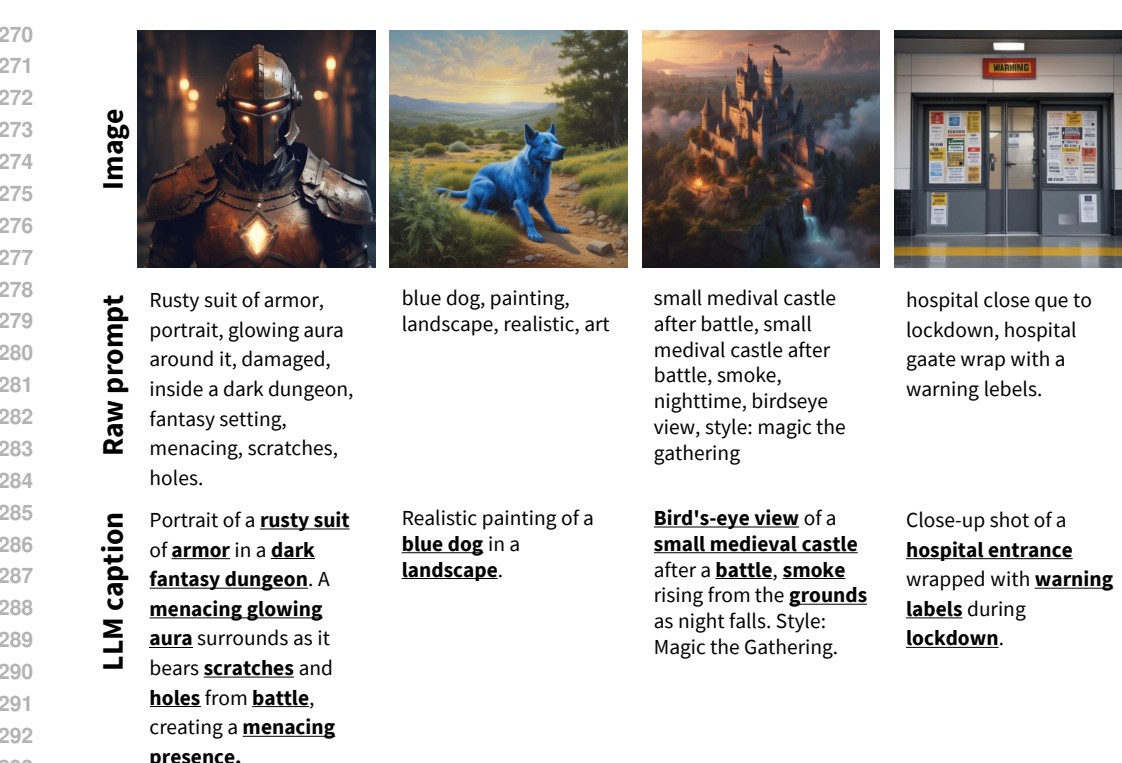

**Raw prompt** (row 1): Rusty suit of armor, portrait, glowing aura around it, damaged, inside a dark dungeon, fantasy setting, menacing, scratches, holes.

(row 2): blue dog, painting, landscape, realistic, art

(row 3): small medival castle after battle, small medival castle after battle, smoke, nighttime, birdseye view, style: magic the gathering

(row 4): hospital close que to lockdown, hospital gaate wrap with a warning lebels.

**LLM caption** (row 1): Portrait of a **rusty suit** of **armor** in a **dark fantasy dungeon**. A **menacing glowing aura** surrounds as it bears **scratches** and **holes** from **battle**, creating a **menacing presence**.

(row 2): Realistic painting of a **blue dog** in a **landscape**.

(row 3): **Bird's-eye view** of a **small medieval castle** after a **battle**, **smoke** rising from the **grounds** as night falls. Style: Magic the Gathering.

(row 4): Close-up shot of a **hospital entrance** wrapped with **warning labels** during **lockdown**.

Figure 4: **Example of SDXL generated images from the captions, raw user prompts and LLM processed captions.** Raw prompts from users often contain typos or take the form a non-natural language tag-like format. We instruct an LLM to transform the prompts into a natural language caption. **Noun chunks** (bolded and underlined) are derived from dependency parsing. Images are generated from the captions, with diffusion attribution maps recorded for the noun chunks.

effective localization, we remove articles and possessive determiners if they are the first word of a noun chunk.

## 4 EXPERIMENTS

We first evaluate the CLIP similarity between the generated images and the captions, and further characterize the safety and length of the captions. We then explore the semantic distribution of the images and the captions using CLIP, and visualize the spatial distribution of objects in a scene using diffusion attribution maps. Finally, we evaluate the performance of captioning and open-set segmentation models on our dataset. These characterizations demonstrate how StableSemantics can be a promising dataset for advancing visual semantic understanding. The data will be released under a CC0 1.0 license.

### 4.1 DATASET CHARACTERIZATION

After deduplication and LLM NSFW filtering, we have 224 thousand natural language captions. We evaluate the similarity of the SDXL-lightning generated images and the captions in Figure 3a using OpenAI's CLIP ViT-B/16 (Radford et al., 2021). We find that the CLIP similarity peaks at $0.34$, which is similar to CLIP scores achieved using SDXL. These scores typically range from $0.2$ to $0.5$ which means that our prompts and images can be interpreted to be semantically very similar given the higher range of scores. We visualize the token length of the captions in Figure 3b. Note that we do not generate images for captions exceeding 77 tokens post-padding. This yields a total of 200k captions which are used for image generation. In Figure 3c, we plot the NSFW scores of the captions used for image generation, as evaluated using the state-of-the-art `Meta Llama Guard 2` model. We define the unsafe categories to the 3 official categories relating to sexual content, and the 1 official category related to hate speech. The scores are the "unsafe" softmax outputs between the "safe" and "unsafe" tokens. We find that the captions used for generation are overwhelmingly safe. In Figure 4, we provide examples of the images, the original human generated prompts which may often contain

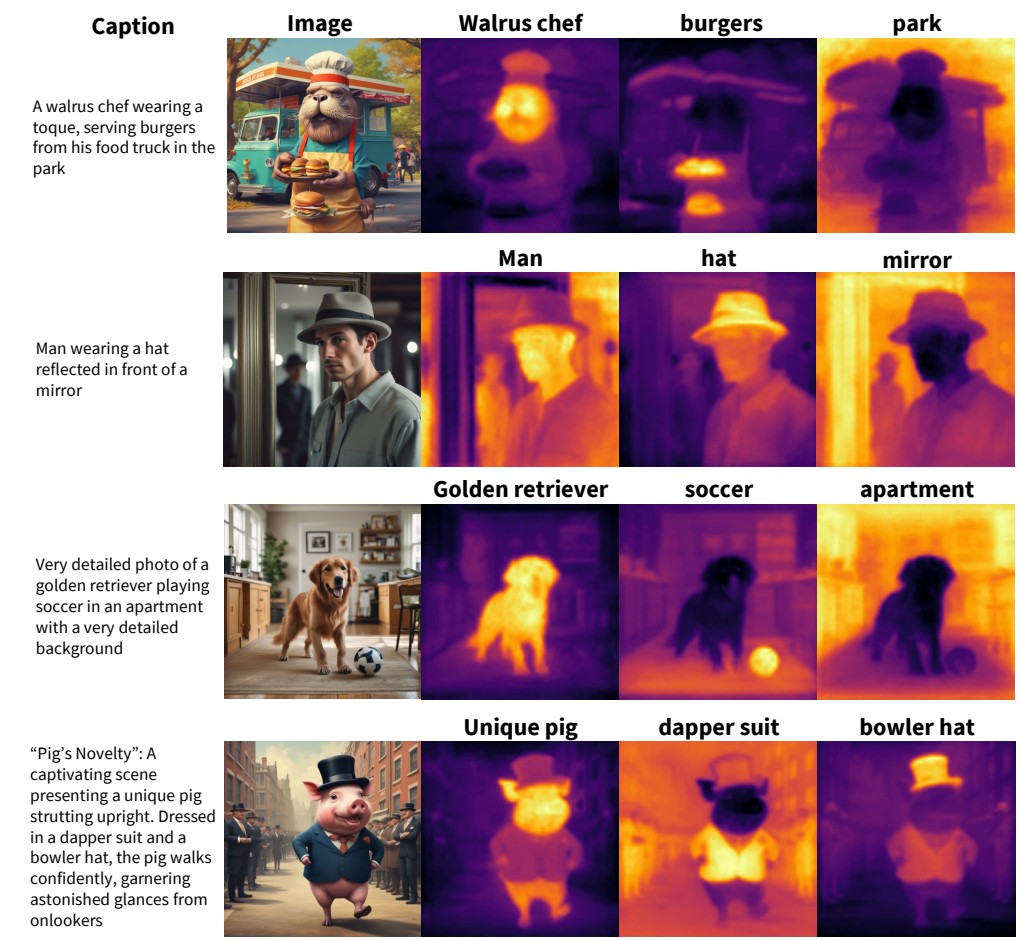

Figure 5: **Visualization of the dataset.** We show example captions used for image generation, images generated from the captions, and select noun chunks and their corresponding attention attribution maps. We find that our dataset contains accurate localizations for different semantic concepts.

typos or tags, and the LLM output natural language prompts. Likely due to human preference, we observe a higher ratio of images with visually interesting compositions.

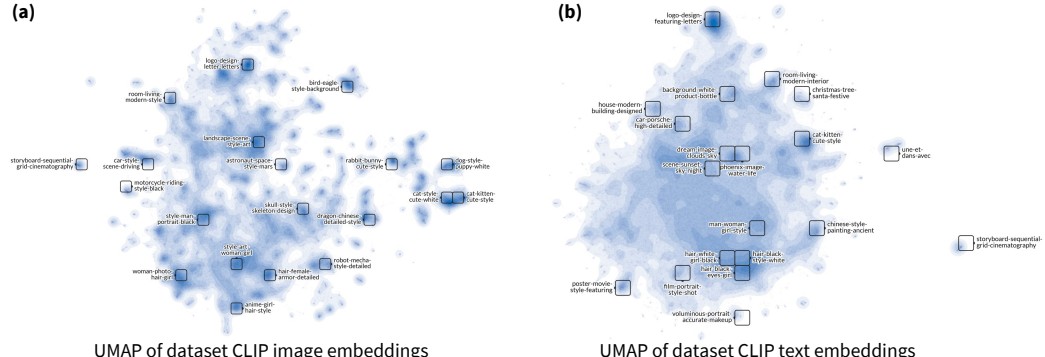

Figure 6: **UMAP visualization of dataset CLIP embeddings.** We use OpenAI CLIP ViT-B/16 to compute embeddings for both the generated **(a)** images and the **(b)** text. UMAP with the cosine metric is used to perform dimensionality reduction. We observe that images describing people, scenes, text, and animals occur with high frequency.

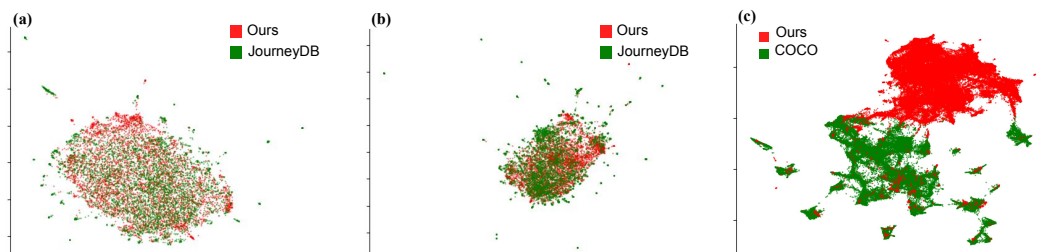

Figure 7: **UMAP visualization of CLIP embeddings of different datasets.** (a) CLIP text embeddings of the natural language captions. (b) CLIP text embeddings of the prompts. (c) CLIP image embeddings of StableSemantics against COCO.

| Model | GPT-2 | Llama 3 8B |
|---|---|---|
| DiffusionDB | 335.86 | 92.02 |
| JourneyDB | 241.39 | 94.85 |
| StableSemantics | 206.26 | 91.41 |

Figure 8: **Perplexity scores for prompts**

| Model | GPT-2 | Llama 3 8B |
|---|---|---|
| JourneyDB | 92.11 | 46.94 |
| StableSemantics | 87.35 | 43.74 |

Figure 9: **Perplexity scores for captions**

In table 8 we evaluate the prompts which are typically not natural language. For JourneyDB and StableSemantics we use the processed prompts (with parameters removed via regex). StableSemantics prompts consistently have lower perplexity than other datasets, likely due to human preference. A lower perplexity score indicates the prompt can be better predicted by the model and is more similar to natural language. Note that we do not use instruct-tuned models. Similarly, in table 9 we evaluate the captions which are LLM-processed prompts. Note that DiffusionDB did not provide captions. Captions from StableSemantics consistently have lower perplexity. This is likely due to using a more powerful LLM (Gemini vs GPT-3.5) and the human preference as a source of the raw captions.

## 4.2 SEMANTIC EXPLORATION OF THE DATASET

In Figure 6 we visualize the semantic distribution of whole images and the captions used to generate the images. We utilize UMAP (McInnes et al., 2018) with a cosine metric applied to CLIP embeddings for this visualization with wizmap (Wang et al., 2023c). We find that the distribution of both images and text exhibit peaks in concepts such as people, scenes, text, and animals (cats and dogs). These peaks likely reflect the effect of human preference on visually interesting images.

We compare the distribution of prompts, captions, and images against JourneyDB and COCO. Figure 7a shows that our captions (natural language outputs of Gemini/GPT) are a subset of JourneyDB's captions. Figure 7b further shows that this cannot be due to the choice of LLM, and instead this is the influence of human preference for visually appealing images as human preferred prompts tile the broader prompt distribution, and do not form a linearly separable set. Figure 7c shows that image distribution of COCO and Our dataset is different and highlights that our dataset has images with objects in complex or imaginative scenarios not observed in real datasets.

The semantic maps we provide in our dataset help localize image regions corresponding to specific noun chunks from the prompts. In Figure 5 we visualize the captions used from image generation, the generated RGB image, and attention attribution masks corresponding to noun chunks shown in bold. We find that our dataset can provide semantic attributions that are well aligned to objects in the scene. This is likely due to the nature of Stable Diffusion, which leverages cross-attention guidance to generate complex compositional images.

We use these maps to analyze whether our images exhibit a trend of certain concepts being generated in specific regions on average. In Figure 10 we aggregate the masks of the top 100 noun chunks that have the highest CLIP similarity scores with concepts of interest. We apply this similarity-based matching to allow for inexact matching. Figure 10 clearly shows that the spatial distribution of concepts can be highly non-uniform. This bias likely reflects the distribution of concepts in natural

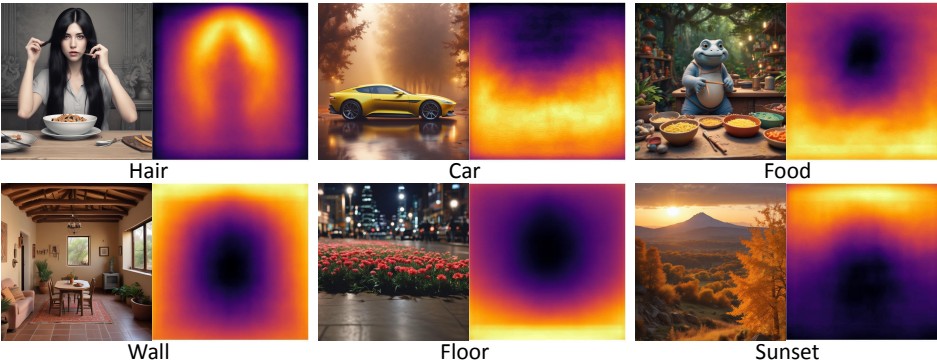

Figure 10: **Spatial distribution of semantic concepts.** For each concept, we visualize an example image containing the concept, as well as the spatial distribution averaged over occurrences. We utilize CLIP text similarity to select the top-100 most similar noun chunks and average those occurrences.

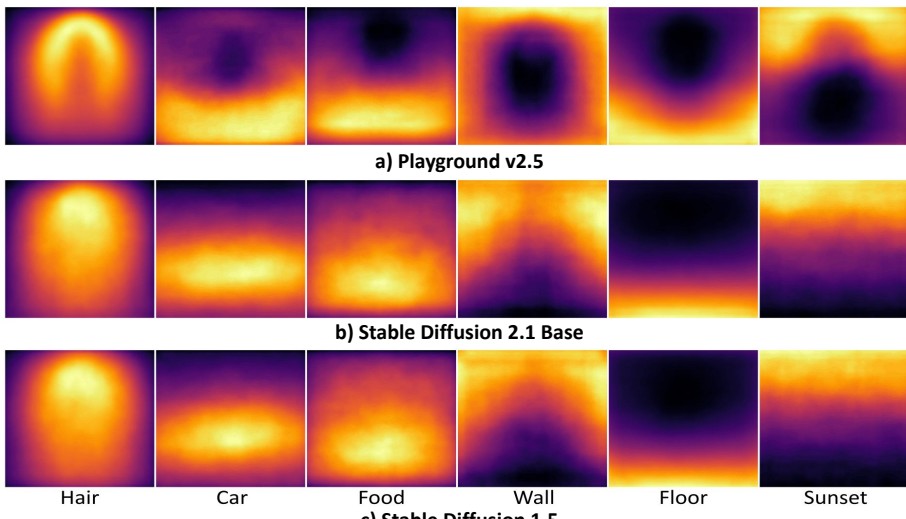

Figure 11: **Spatial distribution of semantic concepts for other generative models.**

images (Torralba & Oliva, 2003; Greene, 2013). For instance, it makes sense for the sunset to always be on the top, the walls towards the sides, and the floor towards the bottom. It is also very common to find human beings as the primary subjects in images which explains the placement of hair surrounding a central region. Finally, many images exhibit food on top of a table and cars on roads. In these scenarios, these semantic concepts typically occupy the bottom half of the visual field. Furthermore, we compare the spatial distribution of concepts across various popular generative models in Figure 11 and find similar trends being followed with slight variations. We visualize the frequency distribution of nouns grouped by wordnet hierarchy in Figure 12. Our dataset could be used to understand the spatial and visual bias present in natural images.

## 4.3 EVALUATION OF MODELS

In this section we evaluate the performance of state-of-the-art open-vocabulary image segmentation and captioning models on our dataset. For open-vocabulary segmentation methods, we evaluate the standard mean Intersection over Union (mIOU), where discrete masks are computed by taking the argmax over all noun chunks' continuous masks for a given prompt. As these methods also produce soft masks, we also evaluate the pearson correlation of the attribution maps from our datasets. In Table 14, We find that recent open-vocabulary segmentation models which modify CLIP (LSeg (Li et al., 2022), SCLIP (Wang et al., 2023a)) or leverage text-to-image diffusion models (ODISE (Xu et al., 2023)) perform better than their peers like MaskCLIP (Dong et al., 2023) CLIPSeg (Lüddecke & Ecker, 2022) and OVSeg (Liang et al., 2023).

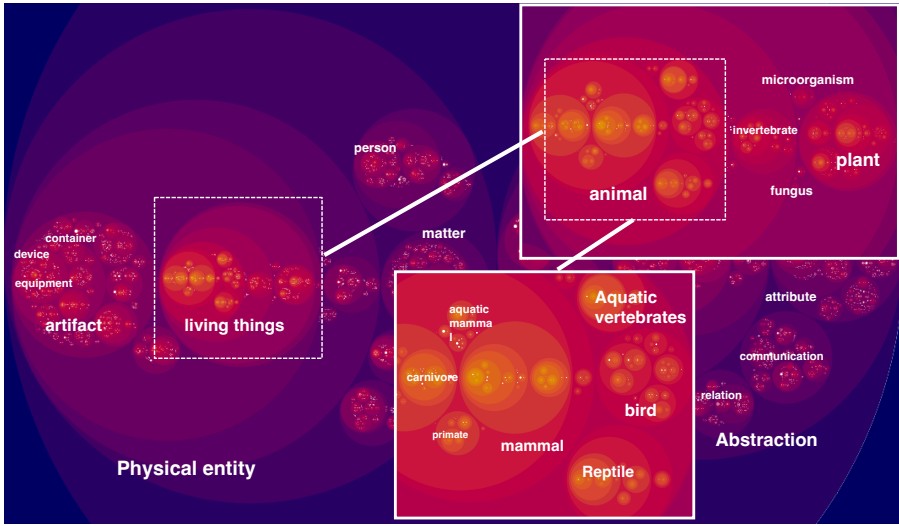

Figure 12: **Frequency of nouns visualized with wordnet hierarchy.** We parse the sentences and extract the nouns. The hierarchy is from wordnet Fellbaum (2010). The circle size corresponds to frequency.

| Methods | E5-Mistral ↑ | BLEU ↑ | CIDEr ↑ |
|---|---|---|---|
| LLaVA | 67.9 | 1.2 | 3.1 |
| BLIP-2 | **70.9** | **1.9** | **10.2** |
| GIT | 63.3 | 1.0 | 6.8 |
| CoCa | 66.8 | 1.7 | 9.7 |

| Method | mIoU ↑ | Pearson ↑ |
|---|---|---|
| MaskCLIP | 0.015 | 0.199 |
| SCLIP | 0.109 | 0.236 |
| LSeg | **0.164** | 0.032 |
| CLIPSeg | 0.133 | 0.143 |
| ODISE | 0.096 | **0.300** |
| OVSeg | 0.035 | 0.181 |

Figure 13: **Performance comparison of captioning models.** We apply captioning models and evaluate the alignment of the outputs against the captions used to generate the images.

Figure 14: **Performance comparison of open-set segmentation models.** We evaluate the IoU and Pearson correlation for noun chunks against model outputs.

We perform an experiment to finetune MaskCLIP (using the ViT-B/16 as backbone, initialized from laion2b_s34b_b88k weights) on a subset of our dataset using a cosine distance loss between the dense predicted output and the argmax-assigned spatial CLIP text embeddings. The training set is disjoint from the test set. Images are only selected for training/testing if they include three or more noun chunks. The Pearson correlation of the finetuned model increases from 0.199 (original MaskCLIP) to 0.38 (finetuned model).

We also evaluate recent vision-language models like LLaVA (Liu et al., 2024), BLIP-2 (Li et al., 2023), GIT (Wang et al., 2022a) and CoCa (Yu et al., 2022). for image captioning in Table 13. We evaluate the generated captions against the original captions using state-of-the-art E5-Mistral (Wang et al., 2023b) model to evaluate cosine similarity (×100 for clarity), BLEU-4 (Papineni et al., 2002), and CIDEr (Vedantam et al., 2015) scores. These results suggest that while the captions predicted by models may use different wording from the original caption, they can be semantically very similar.

## 5 DISCUSSION

**Limitations and Future Work.** Our work relies on human-submitted prompts, which may exhibit non-natural semantic co-occurrences. During the data collection process, we also observed a strong shift in the semantic distribution of prompts and images around holidays (Thanksgiving, Christmas). This suggests that continual data collection is required to mitigate bias.

**Conclusion.** We introduce StableSemantics, the first large-scale dataset that combines natural language captions, synthetic images, and diffusion attribution maps. Our work goes beyond prior datasets by providing spatially localized noun chunk to image region mappings. We explore the semantic distribution of whole images and objects within an image. The availability of this dataset will allow for the use of synthetic visual data in additional domains.

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

# A  APPENDIX

## Sections

## A.1 ADDITIONAL DATASET VISUALIZATION

In this section, we provide additional visualizations of the natural language captions generated using a large language model from the raw user prompts, the RGB image, and semantic masks corresponding to select noun chunks in the image in Figure S.1 and Figure S.2.

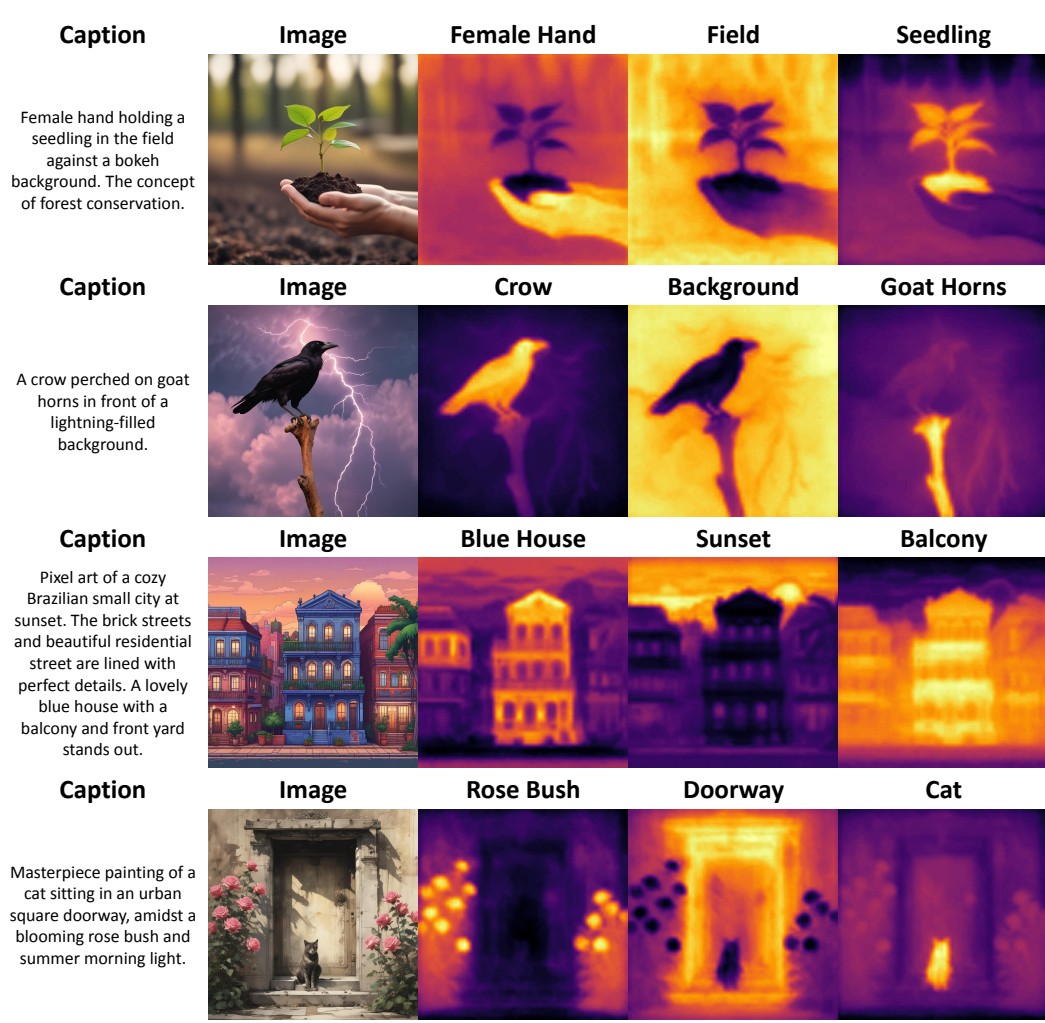

Figure S.1: **Visualization of additional dataset examples.** We show the natural language caption used for the image generation, the image, and masks corresponding to select noun chunks.

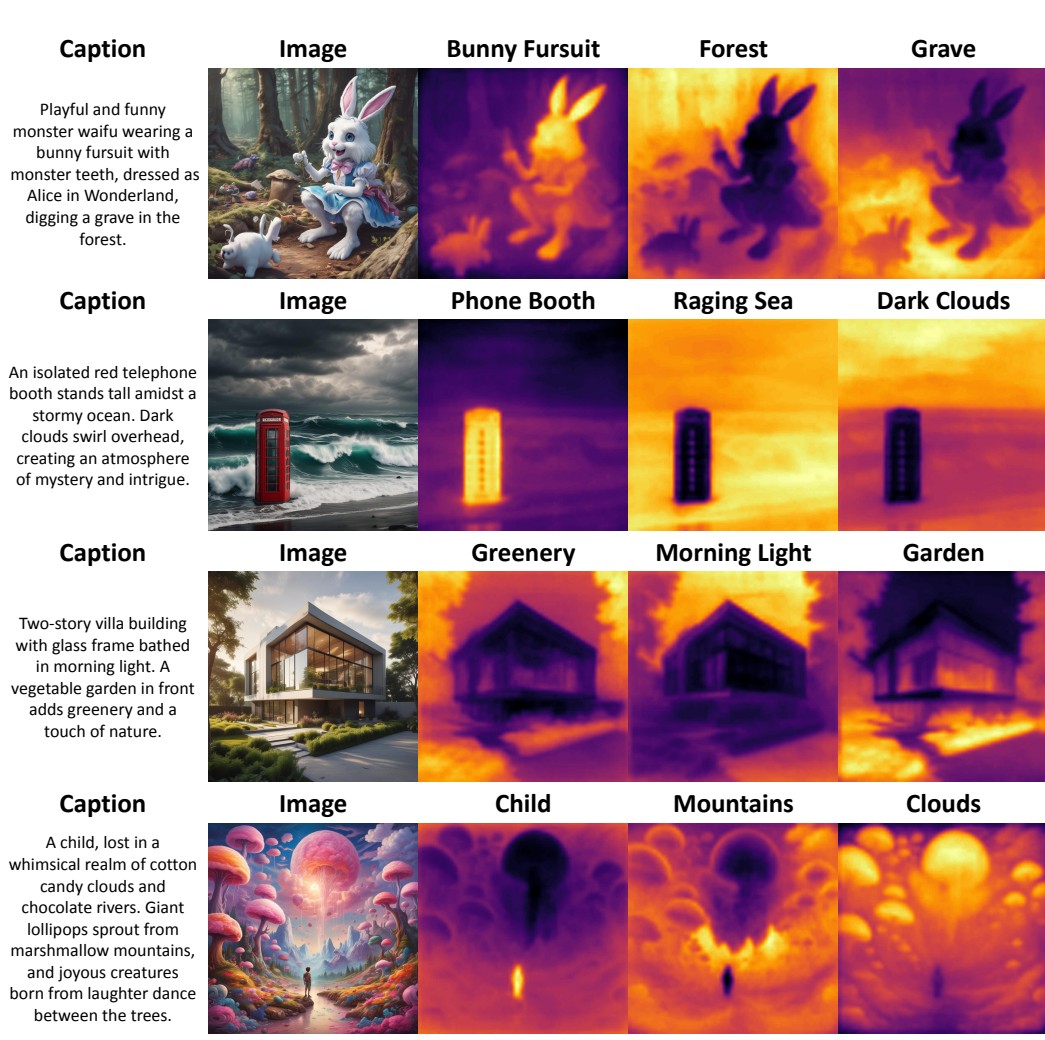

Figure S.2: **Visualization of additional dataset examples.** We show the natural language caption used for the image generation, the image, and masks corresponding to select noun chunks.

## A.2 Visualization of object distributions

In this section, we visualize the spatial distribution of various noun chunks in Figure S.3. We note that several types of objects exhibit highly non-uniform spatial distributions.

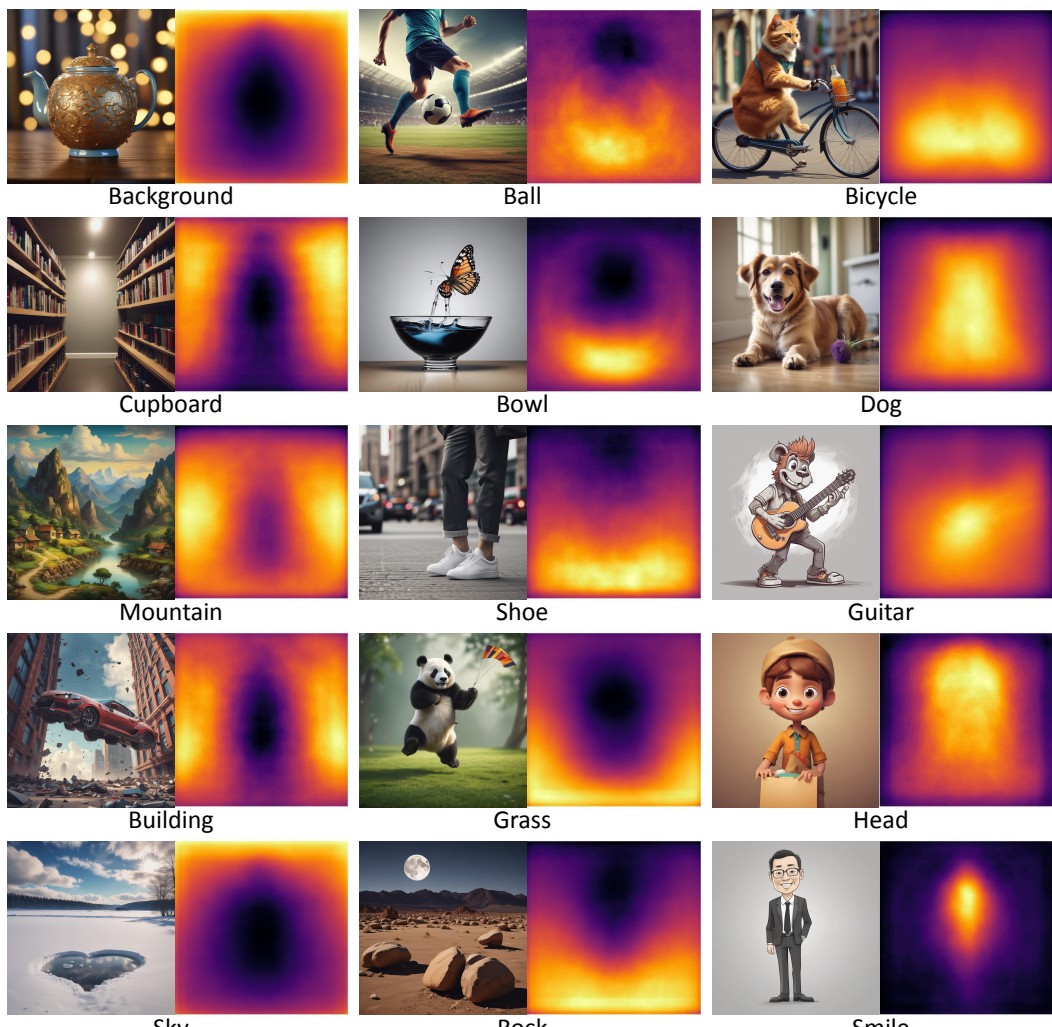

Figure S.3: **Additional examples on the spatial distribution of concepts.** We provide additional examples of images containing a concept, and the average distribution of the top-100 images containing the most similar noun chunks as evaluated using the CLIP text model.

## A.3 COMPARISON OF OPEN VOCABULARY SEGMENTATION METHODS

In this section, we provide additional visualizations of the semantic maps from our dataset, and segmentation outputs in Figure S.4 and Figure S.5. We note that in general, the semantic masks in our dataset can accurately localize objects. However, there are specific cases (Lone Man) where the attention maps corresponding to noun chunks can include other contextual objects. We believe this occurs when the diffusion model tries to generate co-occurring scene and image parts that are not explicitly mentioned in the caption.

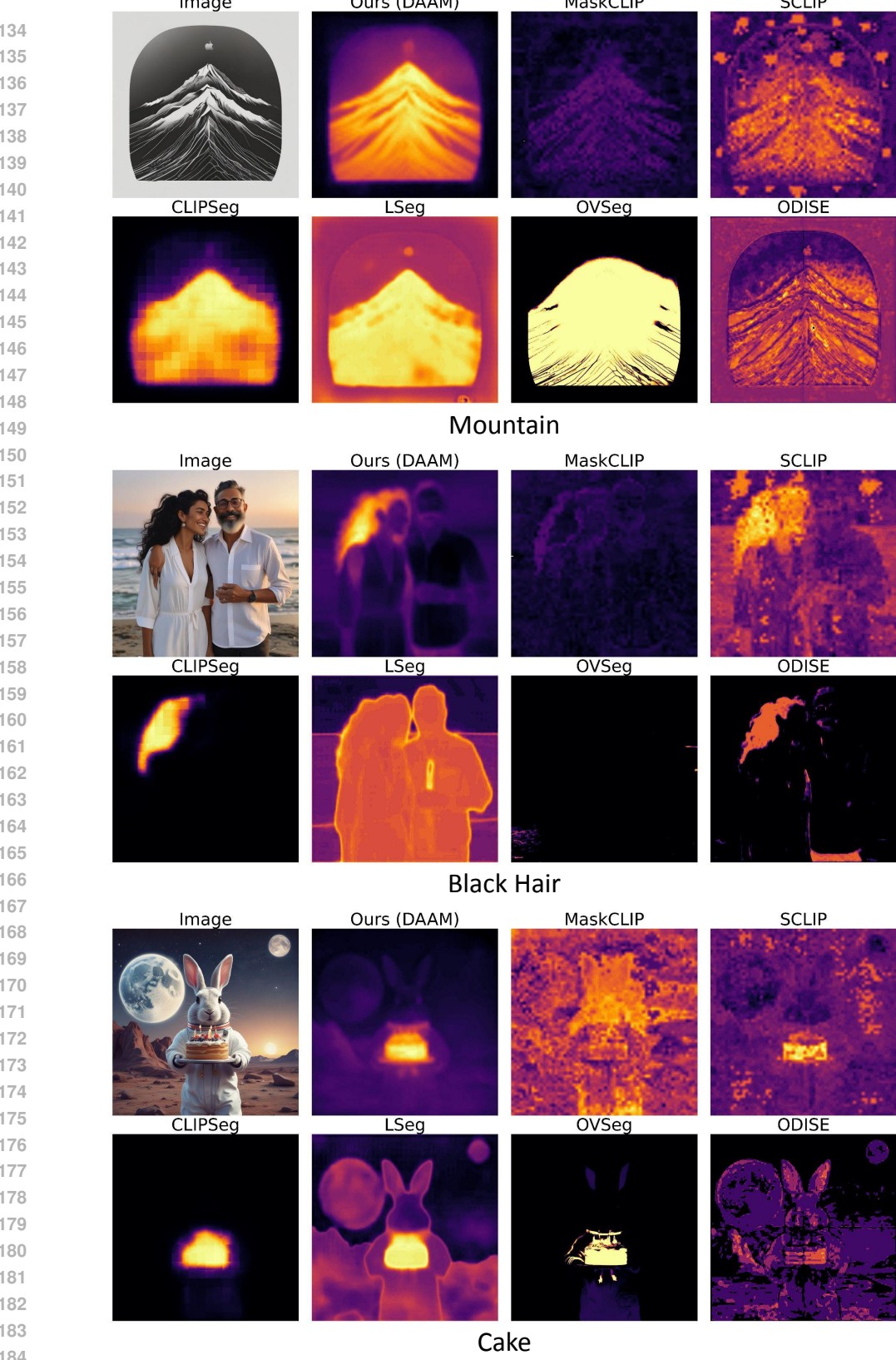

Figure S.4: **Comparison of semantic maps and open vocabulary segmentation methods.** We visualize the semantic attribution maps corresponding to noun chunks from our dataset, and the segmentation maps produced by various state-of-the-art methods.

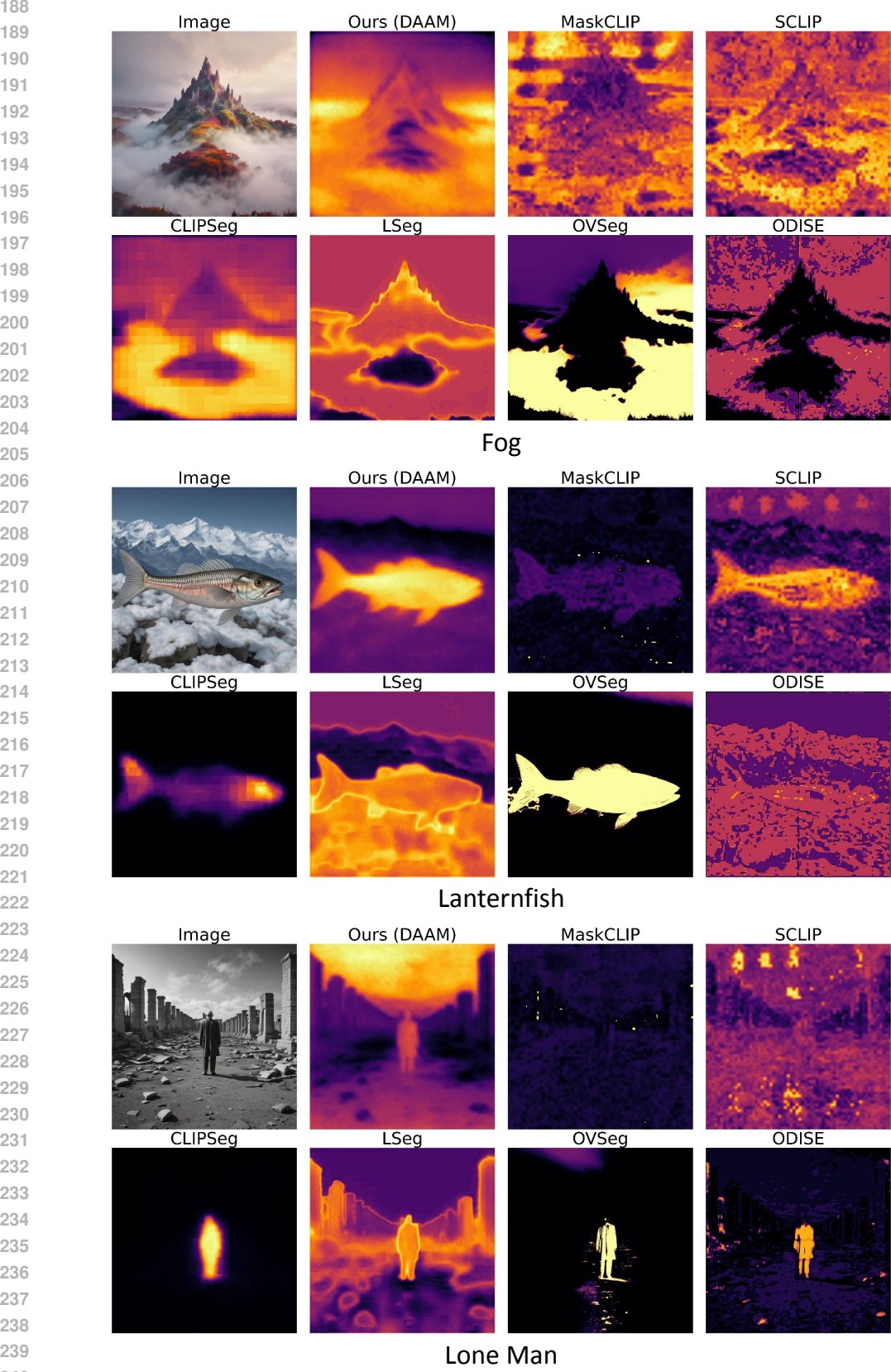

Figure S.5: **Comparison of semantic maps and open vocabulary segmentation methods.** We visualize the semantic attribution maps corresponding to noun chunks from our dataset, and the segmentation maps produced by various state-of-the-art methods. Note that the **Lone Man** illustrates how the semantic attribution maps can be imperfect. In this case it includes additional background.

### A.4    PROMPT USED FOR LANGUAGE MODEL CLEANUP OF RAW PROMPTS

We utilize the following prompt followed by the raw user prompt to obtain a natural language caption. Note that our raw prompts undergo simple `regex` based processing to remove some obvious errors before being provided to the language model.

```
You are going to be provided with the description of an image.
You will transform and edit the description as needed into
a cohesive natural language sentence or sentences without
elaborating.  If the original is mixed language, then your output
should also be mixed language.  Do not elaborate, do not provide
information about people mentioned in the description.

Make a best effort to use ALL WORDS AND DETAILS from the original
description.  DO NOT MAKE UP DETAILS unless absolutely necessary.
BE AS CONCISE AS POSSIBLE, WHILE ATTEMPTING TO INCLUDE ALL WORDS
AND DETAILS FROM THE ORIGINAL DESCRIPTION. You may only omit
details or words if they are nonsensical or form a contradiction.
Attempt to fix typos and remove invalid punctuation.  Omit emojis
in your output.

When you output, use [START] before the output, and include [END]
after the output.  You may retain hash tags in the output only if
hash tags were used in the original prompt.  Here is the original
description:
```

