# OpenReview forum: "StableSemantics: A Synthetic Language-Vision Dataset of Semantic Representations in Naturalistic Images"
_ICLR.cc/2025/Conference — ICLR 2025 Conference Withdrawn Submission_

### Official Review · Reviewer_jttK · 2024-10-21

**Soundness:** 2
**Presentation:** 2
**Contribution:** 2
**Rating:** 3
**Confidence:** 3

**Summary:**

This paper introduces a new large-scale dataset that combines text-to-image generations with semantic annotations, comprising 224K human-curated prompts and 2M synthetic images. The dataset provides fine-grained semantic attributions at the noun-chunk level, along with 10 million attention maps that link textual descriptions to specific image regions. The authors analyze the semantic distribution of generated images and benchmark various vision tasks.

**Strengths:**

1. The paper is easy-to-understand.
2. Captioning and segmentation models are evaluated.
3. The prompts are manually filtered to ensure quality.

**Weaknesses:**

My major concern is the significance of the proposed dataset. The main difference between the proposed dataset and previous counterparts is the additional dense open-set spatial semantic maps. The authors claim that "we expect StableSemantics to catalyze advances in visual semantic understanding and provide a foundation for developing more sophisticated and effective visual models". However, the reviewer remains unclear about the usage of semantic maps. How are semantic maps being used in existing model training (e.g., VLMs, MLLMs)? In addition, whether the proposed dataset can actually benefit downstream tasks remains doubtful. More experiments are needed to demonstrate the effectiveness.

**Questions:**

See weaknesses.

---

### Official Review · Reviewer_dhkL · 2024-11-02

**Soundness:** 3
**Presentation:** 3
**Contribution:** 2
**Rating:** 5
**Confidence:** 4

**Summary:**

This work contributes a large-scale synthetic language-vision dataset called StableSemantics. The dataset is constructed in several stages. First, human-written prompts are curated from the Stable Diffusion discord server in a manner that accounts for user rankings of the resulting image. Each prompt is post-processed (using a combination of regular expressions and LLM re-writing) to form a caption, and this caption is used to synthesize images using a distilled variant of Stable Diffusion XL. Finally, diffusion attribution maps are computed for noun chunks appearing in the caption to generate dense semantic attribution labels.

The key contribution claimed by the authors is the construction of the first large-scale synthetic dataset that includes semantic maps in addition to images and captions. In support of this contribution, the authors conduct various analyses of StableSemantics. These include analysis of the spatial distribution of cross-attention activations corresponding to individual noun chunks, visualization of CLIP embeddings and perplexity measurements from prompts and captions. As an application of the dataset, the authors evaluate open-vocabulary segmentation methods and captioning models on their dataset.

**Strengths:**

Significance: Although there are some limitations to the use of cross-attention maps (see discussion below), I think there is considerable value in constructing synthetic media datasets.  In particular, I agree with the authors’ observation (L088-L089) that generative media has proliferated over social media, advertisements and news media. Consequently, it is important that the community develops methods that work well on such data. I think this direction is relatively undervalued within the Computer Vision community, and this dataset represents a step in this direction.

Originality: As noted by the authors, this dataset appears to be the first large-scale synthetic dataset to contribute semantic maps in addition to images and captions. Although I have concerns about the use of these maps (see discussion below), this is an interesting contribution. If the maps were of high quality, I think this could have been particularly impactful.

Quality: The dataset itself is relatively large-scale (224K curated human-written prompts). This quantity brings a certain quality by enabling more robust statistical analysis of the data than would otherwise have been possible with smaller datasets.

Clarity: Overall, the paper is fairly clearly written.  The figures are nicely captioned and presented. I particularly appreciated the inclusion of failure cases in the appendix (e.g. the “Lone Man” examples in Figure S.5), which helps to give the reader a sense of the robustness of the semantic maps.

**Weaknesses:**

1) One of the key contributions of this work, relative to prior work, is the inclusion of semantic maps. In L095-L096, the authors write: “Specifically, there are no existing large-scale generated datasets that contribute semantic maps in addition to images and their corresponding captions.”  In writing this weakness, I’m interpreting the authors’ claimed contribution to be semantic maps that are similar in usefulness/accuracy to semantic segmentation or instance segmentation annotations.  This interpretation comes from two sources: (1) the listed applications in L111-L112 “StableSemantics can be utilized in future research on various vision tasks such as object detection, semantic segmentation, semantically meaningful representation learning, image-inpainting and object removal, object editing, etc.“, (2) the use of the semantic maps to evaluate open-set segmentation models in Figure 14.

    Unfortunately, relative to traditional instance or semantic segmentation annotations, the semantic maps provided are pretty noisy in capturing the spatial extent of objects or "stuff". As such, it is unclear how useful they are.   A good example is the “mirror” mask in Figure 5, which does not segment the mirror, and instead encodes the surrounding mirror frame.  One consequence of this noise is that it means that downstream evaluations (such as those conducted on open-set segmentation models in Figure 14) don’t seem particularly informative.

   As noted above, I’m listing this point as a weakness on the assumption that the authors intend to frame these semantic maps as being essentially semantic segmentation annotations. It may be that this is a misinterpretation on my part, and they solely wish to claim that they are providing generic semantic attribution maps that don’t aim to capture the spatial extent of objects. This would remove the weakness related to noise, but it brings another problem: it makes the dataset less useful overall.

2) I’m slightly concerned that the generated media is specific to a single model (namely sdxl lightning 4step unet). This may not be an issue - perhaps this model is highly representative of many modern diffusion models. However, it is difficult to ascertain from the paper whether this dataset will immediately go “stale” when new models are released. One possible way to mitigate this would be to generate images from multiple models and demonstrate that similar qualitative insights can be drawn from the data associated with each model.

**Questions:**

1. Did the authors perform any experiments to quantify the quality of the semantic maps, assuming that the authors wish to use them to characterize the spatial extent of objects? For instance, one way to mitigate concerns about quality would be to manually annotate a small number of images and compute mIoU metrics between the manual annotations and the semantic maps.

2. Given the noise issues discussed above, are there other potential applications of the semantic maps that could be useful for the community? One experiment that could demonstrate value would be to show that the authors could train on these maps and improve accuracy on a hand-annotated test set. (This would be similar to the MaskCLIP experiment you conduct in L518, but would change the test set to use trusted annotations.)

3. Using an LLM to rewrite the human prompts seems likely to subtly change the distribution of generated images. Do the authors have any ideas for how you could quantify this distribution shift? (One option might be to characterize differences between images generated by the prompts and images generated by the captions).  To give a specific example, the user asks for a hospital gaate (sic) in Figure 4, but the LLM caption rewriting changes this to “hospital entrance”. This is not necessarily what the user wanted.

4. This is a question rather than a comment since I’m not a lawyer. Do the authors have the rights to release this dataset under a CC0 1.0 license, as proposed in L308-L309? Reading the Stable Diffusion discord terms of service (https://stability.ai/discord-tos, accessed 30th October 2024) my understanding is that Stability owns the content on the server. A CC0 1.0 license would make this discord server content available for downstream commercial use.

5. This is a low importance question (the authors may feel free to ignore it if preferred). In L467, it says “it makes sense for the sunset to always be on the top, the walls towards the sides, and the floor towards the bottom.” However, examining Figure 10, the “Wall” map seems to be strongest at the top of the image. Why do the authors think this is the case?

---

### Official Review · Reviewer_o8p9 · 2024-11-03

**Soundness:** 4
**Presentation:** 3
**Contribution:** 2
**Rating:** 3
**Confidence:** 4

**Summary:**

This paper introduces StableSemantics, a synthetic dataset which consists of the user prompt, the derived caption, the generated image and attention maps for each noun in each prompt. This can be used to find the semantic distribution of images, object relations, and provide a useful benchmark for dense visual semantic tasks.

**Strengths:**

- this paper was clear in its dataset collection process and I particularly appreciated how comprehensive the data analysis was
- the addition of semantic attribution maps could be valuable in providing more fine grained semantic labels
- the dataset collection process as described requires very little human annotation and takes from a live datasource; I can see a promising direction in periodically recollecting data to ensure data freshness

**Weaknesses:**

My main concerns surround the value add of this dataset in comparison to DiffusionDB and JourneyDB
- the authors claim that StableSemantics differs from datasets like JourneyDB because StableSemantics only collects prompts which are human generated. However, from looking through the JourneyDB collection process it seems that they also collect user prompts from discord servers: is the difference that JourneyDB also contains prompts which are not human generated? If so, this seems like a very small delta.
- the authors show in table 8 that their prompts and captions have a lower perplexity score and thus is more similar to human language than DiffusionDB and JourneyDB, but it is unclear to me why this is desired.
- In figure 7 the authors show that the captions are a subset of JourneyDB's, indicating to me that these captions don't capture a different distribution than existing datasets. This seems like an argument that this dataset provides little value
- there is a lot of mention about the distribution of prompts and captions being specifically objects/styles/scenes that users would find visually appealing. While i appreciate the authors honesty, I do think this reduces the value of this dataset as it only captures a small distribution compared to natural images which are often inputs into captioning and segmentation models.

While an interesting idea, I do not think that this paper has shown that it provides substantial value when compared to existing datasets. That being said, I hope the authors can convince me otherwise by giving some applications of when someone would prefer this dataset over the others. One immediate thing I can think of is to generate the attribution maps for all or a subset of JourneyDB and see if the model lineup changes for the captioning and segementation models. If these differ and the authors show that the evaluation on their dataset provides insights into these models that JourneyDB does not, I would be compelled to improve my score. I am also happy to clarify any of my points in the rebuttal if they were unclear

**Questions:**

- [intro] "To overcome this challenge, recent advances have adopted data-driven approaches, which learn to recognize patterns and relationships in large datasets of images and annotations." - what are the works which do this?
- the authors mention that converting the prompts into captions is necessary because the prompts often have misspellings and unnatural sentence structure: why does this matter? is it because it is hard to parse the nouns with incorrect spelling?

---

### Official Review · Reviewer_uPE4 · 2024-11-04

**Soundness:** 1
**Presentation:** 3
**Contribution:** 2
**Rating:** 3
**Confidence:** 4

**Summary:**

The paper presents "StableSemantics," a synthetic dataset aimed at improving visual scene understanding by combining human-curated text prompts, synthetic images, and dense attention maps at the noun-chunk level. The dataset is built using cross-attention mechanisms within diffusion models, theoretically allowing alignment between text prompts and visual details in generated images. It claims to address limitations in existing datasets by offering more nuanced semantic attributions.

**Strengths:**

1- The StableSemantics dataset maps out exactly which parts of synthetic images correspond to specific objects and concepts. It contains 2 million images paired with 10 million attention maps that show where different words and phrases appear in each image. Unlike previous datasets, it provides detailed spatial information about a wide range of objects and concepts.

2- The  StableSemantics is the first diffusion dataset to record spatial distribution of cross-attention activations for individual noun chunks, positioning it as a pioneering resource for exploring spatial semantics in synthetic data​

**Weaknesses:**

1- As noted in recent studies [1,2], text encoders in models like CLIP have difficulty preserving detailed relational attributes between objects. Since Stable Diffusion XL also uses a CLIP encoder, there is a high probability that generated images include all requested objects and attributes but fail to capture accurate spatial or functional relationships. For example, when prompted with “a banana with checkered skin,” the model may produce an image with a banana on a checkered background, misrepresenting the intended attribute. This limitation restricts the model’s ability to generate complex scenes that require precise relationships between elements. Given this limitation, a human evaluation step is necessary to filter out images that do not align precisely with their prompts, ensuring better relational consistency in the dataset.

2- The paper relies on CLIP scores and DAAM attribution maps as primary metrics to assess image quality. However, these metrics primarily capture object presence rather than the accuracy of spatial or relational context. To ensure that each generated image reflects the intended relationships and layout specified by the prompt, relation-specific metrics or additional human assessment are recommended. This would enhance the dataset’s value for training models capable of nuanced scene understanding and relational accuracy.

[1] Yuksekgonul, M., Bianchi, F., Kalluri, P., Jurafsky, D., & Zou, J. (2022). When and why vision-language models behave like bags-of-words, and what to do about it?. arXiv preprint arXiv:2210.01936.

[2]  Hsieh, C. Y., Zhang, J., Ma, Z., Kembhavi, A., & Krishna, R. (2024). Sugarcrepe: Fixing hackable benchmarks for vision-language compositionality. Advances in neural information processing systems, 36.

**Questions:**

What guarantee is there that the generated images contain the relationships specified in the prompt? Even though the prompts have undergone multiple stages of human and automated filtering, without knowing the exact seed used to generate these images, it is impossible to produce images identical to those that received high ratings on Discord. Moreover, neither CLIP score nor DAAM accurately interprets these relationships and can easily make errors. Consequently, human evaluation is still needed to filter out images that do not align with their prompts.

---

### Note · Authors · 2024-11-17

I have read and agree with the venue's withdrawal policy on behalf of myself and my co-authors.